# Lactoferrin Prevents Chronic Alcoholic Injury by Regulating Redox Balance and Lipid Metabolism in Female C57BL/6J Mice

**DOI:** 10.3390/antiox11081508

**Published:** 2022-07-31

**Authors:** De-Ming Li, Yun-Xuan Wu, Zhi-Qiang Hu, Tian-Ci Wang, Li-Li Zhang, Yan Zhou, Xing Tong, Jia-Ying Xu, Li-Qiang Qin

**Affiliations:** 1School of Public Health, Suzhou Medical College of Soochow University, 199 Renai Road, Suzhou 215123, China; dmli@stu.suda.edu.cn (D.-M.L.); 20206947002@stu.suda.edu.cn (Y.-X.W.); 20215247019@stu.suda.edu.cn (Z.-Q.H.); wtc970419@163.com (T.-C.W.); 20214247012@stu.suda.edu.cn (L.-L.Z.); 20215247044@stu.suda.edu.cn (Y.Z.); 2Laboratory Center, Suzhou Medical College of Soochow University, 199 Renai Road, Suzhou 215123, China; tongxing@suda.edu.cn; 3State Key Laboratory of Radiation Medicine and Protection, School of Radiation Medicine and Protection, Suzhou Medical College of Soochow University, 199 Renai Road, Suzhou 215123, China

**Keywords:** lactoferrin, chronic alcoholic liver injury, redox balance, lipid metabolism, female mice

## Abstract

This study aimed to investigate the preventive effects of lactoferrin (Lf) on chronic alcoholic liver injury (ALI) in female mice. Female C57BL/6J mice were randomly divided into four groups: control group (CON), ethanol administration group (EtOH), low-dose Lf treatment group (LLf), and high-dose Lf group (HLf). In the last three groups, chronic ALI was induced by administering 20% ethanol ad libitum for 12 weeks. Mice in the CON and EtOH groups were fed with AIN-93G diet. Meanwhile, 0.4% and 4% casein in the AIN-93G diet were replaced by Lf as the diets of LLf and HLf groups, respectively. HLf significantly reduced hepatic triglyceride content and improved pathological morphology. HLf could inhibit cytochrome P450 2E1 overexpression and promote alcohol dehydrogenase-1 expression. HLf activated protein kinase B and AMP-activated protein kinase (AMPK), as well as upregulating nuclear-factor-erythroid-2-related factor-2 expression to elevate hepatic antioxidative enzyme activities. AMPK activation also benefited hepatic lipid metabolism. Meanwhile, HLf had no obvious beneficial effects on gut microbiota. In summary, Lf could alleviate chronic ALI in female mice, which was associated with redox balance and lipid metabolism regulation.

## 1. Introduction

Alcohol-associated liver disease (AALD) is a serious public health issue worldwide [1]. Due to a lack of effective pharmacological therapy, the AALD prevention by diet or natural agents is a plausible strategy [2]. Lactoferrin (Lf) is a natural protein in milk with various biological activities that may be relevant for improving AALD [3,4]. Our study also has confirmed the preventive effects of Lf on alcoholic liver injury (ALI) in male mice [5]. 

In most cases, females are more susceptible to alcohol than males [6]. However, less attention has been given to studies on in females. In our previous studies, we focused on the effects of Lf on acute ALI in female mice, and the results also showed the protective effects of Lf [7]. However, acute and chronic alcohol exposure patterns do not share identical pathological process, even show an opposite effect on some specific signaling pathways [8]. Whether Lf treatment can improve liver injury induced by chronic ethanol consumption in female mice still deserves our research.

At low alcohol concentrations, alcohol dehydrogenase (ADH) is mainly responsible for alcohol oxidation, and cytochrome P450 2E1 (CYP2E1) may only account for 10% of the hepatic total alcohol-oxidizing capacity [9,10]. After binge or chronic alcohol consumption, CYP2E1 metabolism of alcohol can increase alcohol oxidation, and the proportion of CYP2E1 increases greatly in the total alcohol-oxidizing capacity [9]. Ethanol conversion catalyzed by CYPE21 is an important source of excessive reactive oxygen species (ROS) that can trigger oxidative stress and eventually lead to cellular injury [9,11,12]. Our previous study reported that a daily dose of Lf treatment could inhibit CYP2E1 overexpression, reduce ROS production, and prevent ALI in male mice [5]. Another study we conducted found that although CYP2E1 was not affected by Lf, a higher dose of Lf treatment could alleviate acute ALI via improving redox response capacity in female mice [7]. The studies shed light on the fact that it might be discrepant for the specific regulatory mechanisms, but redox balance regulatory should be a key for the preventive effect of Lf on ALI with different types.

AALD has been linked to gut microbiota changes [13,14,15]. Many studies also have found that gut microbiota is a medium of Lf to achieve some biological functions [16,17]. Our study has indicated that gut microbiota plays a supporting role in the protective effects of Lf against ALI in male mice [5]. However, like alcohol sensitivity, gut microbiota also has sexual dimorphism [18,19,20]. It is still unknown as to how long-term alcohol drinking and Lf supplement modulate gut microbiota in female mice.

Thus, as a companion study to our previous studies and to understand more comprehensively the preventive effect of Lf on different patterns of ALI in different genders, we conducted a new animal experiment to investigate the roles of Lf in chronic ALI in female mice. In addition, the potential mechanisms were also explored from the angles of hepatic alcohol metabolism, hepatic redox balance, and gut microbiota.

## 2. Materials and Methods

### 2.1. Reagents

Native bovine Lf (iron saturation 12%) was purchased from Hilmar Cheese Company (Delhi, CA, USA). Ethanol (guaranteed reagent) was purchased from Chinasun Specialty Products Company (Suzhou, China).

### 2.2. Animals and Treatments

Female 6–8-week-old C57BL/6J mice were obtained from Jihui Laboratory Animal Care Company (Shanghai, China). The animals were housed in a standardization SPF animal laboratory under a 12 h light–dark cycle. After 1 week of acclimation, they were randomly divided into 4 groups and fed with different diets: (1) control group (CON, *n*=10): AIN-93G diet; (2) ethanol administration group (EtOH, *n*=12): AIN-93G diet; (3) low-dose Lf group (LLf, *n*=12): AIN-93G diet with 0.4% casein replaced by Lf; (4) high-dose Lf group (HLf, *n*=12): AIN-93G diet with 4% casein replaced by Lf. The Lf dose selection reasons have been demonstrated in a previous study [5]. The diet compositions are shown in Appendix A. The Modeling methods of chronic ALI are shown in Figure 1A. Mice in EtOH, LLf, and HLf groups were given 10% (*v*/*v*) EtOH for 3 days and 15% (*v*/*v*) EtOH for 4 days to adapt EtOH in drinking water. Then, the mice were placed on 20% (*v*/*v*) EtOH and maintained at this concentration for 12 weeks to induce chronic ALI. The mice in the CON group received regular drinking water, and all mice had access to food and water ad libitum. The weights of the mice and the consumptions of food and drinking liquid were recorded weekly.

The experiment was approved and supervised by the Soochow University Animal Ethics Committee (approval number: 202009A661) and performed in accordance with the National Institutes of Health’s Guide for the Care and Use of Laboratory Animals.

### 2.3. Sample Collection

The feces were collected and stored in liquid nitrogen on the third day before the end of the experiment. After overnight fasting, all the animals were weighed, anesthetized, and sacrificed. Blood was collected, and serum samples were obtained by centrifugating at 3000 rpm for 10 min, and then stored at −80 °C. The livers were quickly dissected from the mouse body, weighed, quickly frozen in liquid nitrogen, and then stored in an ultra-low temperature freezer for further analyses.

### 2.4. Hepatic Triglyceride (TG) Content and Serum Transaminase and Carbohydrate-Deficient Transferrin (CDT) Level Determinations

Hepatic triglyceride contents were determined by a commercial kit (Jiancheng, Nanjing, China). Serum transaminase levels, including alanine aminotransferase (ALT) and aspartate aminotransferase (AST), were determined by the corresponding kits (Solarbio, Beijing, China) according to the technical manuals. Serum CDT contents were determined by an ELISA kit (Animal Union, Shanghai, China).

### 2.5. Hepatic Histological Analysis

The liver samples were fixed in 4% paraformaldehyde for 24 h, and then the samples were sent to Sevicebio Technology Company (Wuhan, China) for histological section manufacture. The detailed methods are shown in Appendix A.

### 2.6. Hepatic Antioxidase and Malondialdehyde (MDA) Level Determination

The liver samples were homogenized with the ratio of liver weight (mg) to lysate (μL) of 1:9. The homogenates were centrifuged, and the supernatants were collected. Hepatic superoxide dismutase (SOD) and catalase (CAT) activities were determined by the corresponding kits (Beyotime, Shanghai, China). Relative activity was standardized using EtOH group as a reference. Hepatic MDA contents were determined using the kit (Beyotime, Shanghai, China) according to the user’s instruction.

### 2.7. Western Blots

Total proteins from the liver samples were extracted with ice-cold RIPA lysis containing protease inhibitors and phosphatase inhibitors (Beyotime, Shanghai, China), and the lysates were centrifuged at 4 °C to collect the supernatants. Then, the supernatants were mixed with 5× dual color protein loading buffer (Fudebio, Hangzhou, China) and boiled at 98 °C for 10 min. An equal amount of protein (30 μg/lane) was separated on an SDS-PAGE gel. Then, the proteins were transferred onto a PVDF membrane. After blocking and incubating with the primary antibodies and the secondary antibodies, the membranes were developed with Femto ECL reagent (Fudebio, Hangzhou, China). The relative protein expression levels were analyzed using Gel-Pro Analyzer software (Media Cybernetics, Rockville, MD, USA) with GAPDH or vinculin as an internal control. The primary antibody information was summarized in Table 1.

### 2.8. 16S rDNA Sequencing

The feces samples of mice were sent to LC Bio (Hangzhou, China) for 16S rDNA sequencing. The detailed methods are shown in Appendix A.

### 2.9. Statistical Analyses

All data were displayed as “mean ± standard error (SE)” unless specified otherwise. One-way ANOVA with Tukey’s multiple comparison test was used to test the differences among groups. A *p*-value less than 0.05 was considered to be statistically significant. All analyses were performed using SPSS version 21.0 (IBM, New York, NY, USA). Figures were plotted by GrapahPad Prism 8.0 (GraphPad Software Inc., San Diego, CA, USA) and OriginPro 2021 (OriginLab, Northampton, MA, USA).

## 3. Results

### 3.1. Effects of Lf Treatment on Basic Profiles

To assess the effects of Lf on energy intake and weight, we recorded the food and liquid consumptions and measured weekly the weights of the mice. There was no difference in alcohol intake among the three ethanol administration groups. Although EtOH administration led to a reduction in food consumption, the total energy intake slightly increased due to drinking EtOH-containing liquid. (Figure 1B). Serum CDT level is a biomarker for evaluating alcohol consumption [21]. Compared with the CON group, serum CDT contents were significantly elevated in EtOH, LLf, and HLf groups, whereas there was no difference among the three groups (Figure 1C). The body weights increased steadily during the experiment, and no between-group difference was observed (Figure 1D). There was no significant difference in fasting glucose levels among the four groups (Figure 1E). Additionally, EtOH administration increased liver weight, and the increased liver weight was not affected by Lf treatment (Figure 1F). The mice appeared “barbering” in EtOH and HLf groups; however, the phenomenon was almost completely reversed in the LLf group (Appendix A).


Figure 1Modeling methods and basic indices of the mice. (**A**) Modeling methods. (**B**) Food, liquid, and energy intake of the mice. (**C**) Serum carbohydrate-deficient transferrin contents. (**D**) Changes in body weights. (**E**) Fasting glucose of the mice. (**F**) Liver weights of the mice. CON, control group; EtOH, ethanol administration group; LLf, low-dose lactoferrin group; and HLf, high-dose lactoferrin group. Data are displayed as “mean ± SE”. *n* = 10, for CON group; *n* = 12, for EtOH, LLf, and HLf groups. # EtOH vs. CON, *p* < 0.05.
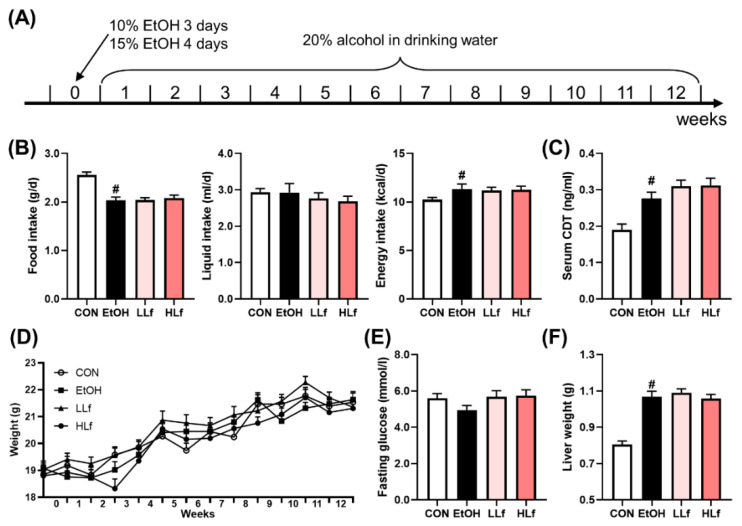



### 3.2. Effects of Lf Treatment on Ethanol-Induced Liver Injury

Alcohol exposure led to a significant hepatic triglyceride accumulation, while Lf-treatment reduced hepatic TG content in a dose-dependent manner (Figure 2A). Long-term alcohol administration also significantly elevated serum ALT levels, which were not affected by Lf treatment. Although AST levels increased slightly in the EtOH group and HLf tended to decrease it, the differences were insignificant (Figure 2B). Hepatic histological morphology is displayed in Figure 2C. Obvious necrosis was observed in EtOH and LLf groups, while the necrosis degree was remarkably reduced in the HLf group. HLf reduced the number and area of lipid vacuolations induced by long-term alcohol exposure to a greater extent than LLf.

### 3.3. Effects of Lf Treatment on Alcohol Metabolism Key Enzymes in Liver

As shown in Figure 3, Lf treatment promoted ADH1 protein expression in a dose-dependent manner. Long-term alcohol administration induced hepatic CYP2E1 protein overexpression, while Lf could inhibit the overexpression. Meanwhile, neither alcohol administration nor Lf treatment affected hepatic ALDH2 protein levels.

### 3.4. Effects of Lf Treatment on Hepatic Redox-Sensitive Proteins

Ethanol exposure inhibited ERK1/2 phosphorylation without affecting total ERK1/2 protein expression (Figure 4A). Further, no significant difference was observed in total AKT and AMPK protein levels. The inhibition of hepatic AKT and AMPK phosphorylation levels induced by long-term ethanol administration could only be restored with HLf but not by LLf (Figure 4B).

### 3.5. Effects of Lf Treatment on Hepatic Redox Homeostasis Regulatory Ability

As shown in Figure 5A, long-term ethanol intake induced a significant decrease in Nrf2 protein expression levels without affecting Keap1 protein level, and Lf treatment was able to restore hepatic Nrf2 protein expression. There were no differences in SOD1 and CAT protein expression levels among the four groups (Figure 5B). Although no difference was observed in hepatic SOD and CAT activities among CON, EtOH, and LLf groups, HLf treatment increased SOD and CAT activities (Figure 5C). Meanwhile, Lf inhibited ethanol-induced hepatic MDA accumulation in a dose-dependent manner (Figure 5D).

### 3.6. Effects of Lf Treatment on Hepatic Lipid Metabolism Key Proteins

ACC and FAS are fatty acid synthesis key proteins. As shown in Figure 6, the up-regulation of ACC and FAS protein expressions and reduction in ACC phosphorylation induced by ethanol administration were restored by Lf, with HLf, showing a stronger effect. CPT1A and HSL play an important role in steatolysis. Compared with the EtOH group, HLf but not LLf could increase CPT1A and HSL protein levels.

### 3.7. Effects of Lf Treatment on Gut Microbiota

Figure 7 shows the rarefaction curves of gut microbiota. The rarefaction curves tended to be flat with the increase in the number of sequences, indicating that the sequencing results were reliable. Although there were no differences in the alpha-diversity indices including observed operational taxonomic units (OTUs), Shannon index, Chao1 index, and Simpson index among CON, EtOH, and HLf groups, the significant reductions in these indices were observed in the LLf group (Figure 7B). Both the principal component analysis (PCA) plot and the hierarchical clustering tree clearly separated the gut microbiota composition of the LLf group from the other three groups (Figure 7C,D).

At the phylum level, the major bacteria were Bacteroidetes and Firmicutes; meanwhile, an increased Verrucomicrobiota relative abundance and a reduced Firmicutes relative abundance were features of gut microbiota composition in the LLf group (Figure 8A). Moreover, at the genus level, *Akkermansia* and *Bilophila* relative abundances were increased but *Eisenbergiella* relative abundance was decreased in the LLf group compared with the EtOH group (Figure 8B). Between-group differential analysis results are shown in Figure 8C. EtOH administration could significantly increase the relative abundances of *Eisenbergiella* and Bacteroides, as well as reduce the relative abundances of *Alistipes*, *Lachnospiraceae*, Escherichia–Shigella, and *Allobaculum*. Both LLf and HLf inhibited *Muribaculaceae* growth and promoted *Alistipes* and *Allobaculum* growth. Interestingly, Lf treatment seemed to have a duality for *Akkermansia* and *Eisenbergiella*: compared with the EtOH group, *Akkermansia* relative abundance was significantly increased by LLf treatment, but declined by HLf treatment, while *Eisenbergiella* relative abundance was just the opposite. In addition, LLf treatment had a significant effect on *Bilophila* and Bacteroides, but HLf treatment did not. 

## 4. Discussion

In this study, alcohol intake was no different for the mice in EtOH administration groups, which was further supported by serum CDT content (a classic biomarker for alcohol consumption [21]) determination results. These indicated that alcohol exposure degree was comparable for the mice in EtOH, LLf, and HLf groups. Long-term alcohol administration increased liver weights, hepatic TG contents, and serum ALT levels. Although HLf treatment did not significantly affect serum transaminase levels, it significantly decreased hepatic TG contents and improved obviously hepatic histomorphological structure, which confirmed the beneficial effects of HLf on chronic ALI in female mice. However, compared with HLf treatment, the beneficial effects of LLf appeared to be extremely limited.

Ethanol can be oxidized to acetaldehyde by ADH in the liver. With the enhancement of dose or prolonged drinking time, cytochrome P450 pathways, especially CYP2E1, are induced to remove alcohol [9,10,22]. Compared with the ADH pathway, the oxidation of alcohol by CYP2E1 usually produces more severe side effects [12,23]. In this study, Lf treatment promoted hepatic ADH1 protein expression and inhibited alcohol-induced CYP2E1 overexpression. The findings indicated that Lf treatment could increase the proportion of ADH pathway in alcohol metabolism to reduce the adverse effects produced by the secondary pathways. Although a direct comparison could not be conducted because our previous study of male mice used a different modeling method, two studies found the suppressive effects of Lf on CYP2E1 overexpression [5]. Acetaldehyde is further oxidized to acetic acid by ALDH2 in the liver [9,24]. Unlike acute alcohol exposure [7], Lf did not affect ALDH2 protein expression in female mice intervened by chronic alcohol intake, which suggested that hepatic ALDH2 expression might not be a key of the preventive effects of Lf on chronic ALI. Meanwhile, because ethanol metabolism key enzyme determinations require sacrificing the animal, a dynamic detection was not conducted in this study. However, it will still be valuable to perform a dynamic measurement of these enzymes at different time points in future studies.

ROS produced from alcohol oxidation can lead to hepatic redox imbalance, which is an important reason for alcoholic liver injury [25]. ERK1/2, AKT, and AMPK signaling pathways are redox-sensitive signaling pathways and play a critical role in sensing redox state and maintaining redox balance [26]. Unlike acute alcoholic liver injury, our present study found that ERK1/2 activation was inhibited in chronic alcoholic liver injury [27]. These findings were consistent with a previous report [28]. Chronic alcohol administration suppressed AKT and AMPK activations without affecting their total protein level, regardless of statistical significance. HLf but not LLf could restore AKT and AMPK phosphorylation, suggesting the critical role of AKT and AMPK in HLf-mediated chronic alcoholic liver injury alleviation. Combined with our previous study on acute alcoholic liver injury [7], we think that the alleviative effects of HLf on alcoholic liver injury in female mice is dependent on AKT and AMPK signaling pathways instead of ERK1/2 signaling pathway. It was worth noting that the mice were sacrificed after fasting, which is considered a factor in activating ERK1/2, AKT, and AMPK signaling pathways [27,29,30]. In this study, chronic ethanol exposure inhibited these protein activations even after fasting. However, HLf treatment reversed the blocked activations of AKT and AMPK. This further demonstrated the critical role of the AKT and AMPK signaling pathways in the process of HLf preventing chronic ALI.

Keap1-Nrf2 signaling pathway is a master regulator of the antioxidant system [31]. Keap1 is a negative regulator of Nrf2, and it can degrade Nrf2. Under oxidative stress, Keap1 undergoes a conformational change so that Nrf2 is dissociated to modulate redox balance via the antioxidase system, including SOD and CAT [32]. Although Keap1 protein levels were not different among the four groups, Lf treatment could reverse alcohol-induced Nrf2 protein expression reduction, contributing to the enhanced antioxidant ability. Alcohol exposure can lead to ROS overproduction [33], and the antioxidase is responsible for ROS clearance [34,35]. Although Nrf2 was increased in LLf and HLf groups, SOD1 and CAT protein expression levels were not affected, and only HLf but not LLf could elevate SOD and CAT activities. We speculated that antioxidant enzymes could work more efficiently in the HLf group than in EtOH and LLf groups. Lf treatment could dose-dependently reduce hepatic MDA accumulation, which further supported the antioxidative role of Lf. Lipid metabolism dysregulation also is an important cause of chronic ALI [36]. We observed a disruption of fatty acid synthesis by ethanol, characterized by the significant upregulation of ACC and FAS protein expressions. Meanwhile, ethanol administration also decreased fatty acid β-oxidation key enzyme CPT1A protein level. Lf treatment inhibited the overexpression of ACC and FAS, but only HLf treatment could upregulate CPT1A and HSL expressions. These findings suggested that the required dosage of Lf in regulating lipolysis was far higher than that required for fatty acid synthesis. Meanwhile, lipid metabolism regulation of HLf likely was derived from the activation of AMPK, since these lipid metabolism key proteins are generally considered to be the downstream proteins of AMPK [37,38]. Moreover, AMPK activation can phosphorylate ACC to inhibit its enzymatic activity [39]. Our study also found both AMPK phosphorylation and ACC phosphorylation were significantly increased in the HLf group compared with the EtOH group. These further confirmed our speculations.

Unabsorbed Lf digestive products can reach the colon and affect gut microbiota [40,41]. Considering the involvement of gut microbiota in ALI, we collected the mouse feces and conducted 16S rDNA sequencing. Overall, LLf showed a more obvious modulation to gut microbiota than HLf. By comparison, HLf had a stronger preventive effect on chronic ALI, which suggested the regulation of liver itself rather than gut microbiota might play a leading role in Lf-mediated liver protection. 

The present study did not find that long-term alcohol intake affected alpha diversity of gut microbiota. Similar to our previous study of male mice [5], LLf treatment significantly reduced the gut microbiota alpha diversity. However, the reduction disappeared in the HLf group. *Akkermansia* has recently been regarded as one beneficial microbe with various metabolic benefits [42,43], and our previous study also emphasized its role in the protective effects of LLf on ALI [5]. In the current study, the *Akkermansia* relative abundance was significantly increased in the LLf group, but significantly decreased in the HLf group. Meanwhile, in our previous study of male mice [5], although both LLf and HLf enhanced *Akkermansia* abundance, the increasing effect was more obvious in the LLf group. The results suggested that higher Lf dose for gut microbiota may not produce more benefits, even leading to deleterious influence. Additionally, approximately half of mice the showed “barbering” in EtOH and HLf groups, but this phenomenon almost completely disappeared in the LLf group. “Barbering” in rodent animals is usually regarded as a neural behavior that may be related to stress [44,45]. The findings also suggested that although HLf had more potent protective effects on chronic ALI than LLf, it might exert a negative influence to other organs or systems. Meanwhile, the potential adverse effects of excessive Lf intervention also have been reported [46]. Thus, we should not only pay attention to the needs of specific disease prevention, but also assess the risk and benefit of other organs and systems, when supplementing Lf.

Lf contains iron [4]; iron overload is associated with ALI and harms hepatic health [47]. Thus, it is possible that Lf supplement could increase iron overload risk. In fact, Lf has a very low iron content (less than 0.02% *w*/*w*), and the iron content of the diet in the HLf group was only 5% higher than that in the CON or EtOH group. We believe that the iron dose was too low to increase the risk of iron overload. Our previous study also confirmed that no hepatic iron accumulation occurred in the female mice fed with the HLf diet for 4 weeks [7].

This study has some limitations. Firstly, we found the obvious protective effects of HLf on chronic ALI, but the dose (approximately equal to an adult ingesting 5000 mL milk per day) was impossible to achieve via a regular diet. Although dietary supplements containing Lf could be a plausible strategy, exorbitant dose and high cost limit the practical application of Lf to a certain extent. Secondly, HLf might cause some side effects while bringing benefits to the liver. Our present study did not determine the optimum dose of Lf supplement. Thirdly, the development of chronic ALI is a multifactorial and complex pathological process [48,49], and Lf as a natural nutrient also affects metabolism in a multifaceted and miscellaneous manner. Therefore, we were only able to explore the potential mechanisms from a limited dimension. It was likely that other important mechanisms were not found, which should be solved in the future.

All in all, the present study found that HLf treatment can alleviate ALI induced by long-term ethanol intake. The potential mechanisms are shown in Figure 9. On the one hand, HLf treatment can suppress CYP2E1 overexpression to minimize oxidative stress. On the other hand, HLf treatment can activate AMPK and AKT signaling pathways, upregulate Nrf2 expression, and improve oxidative-stress-responding capacity. Meanwhile, AMPK-mediated lipid metabolism regulation may play a critical role in hepatic steatosis amelioration. However, the potential side effects caused by excessive Lf intake should not be ignored. Lf supplementation without professional guidance is not recommended.

## Figures and Tables

**Figure 2 antioxidants-11-01508-f002:**
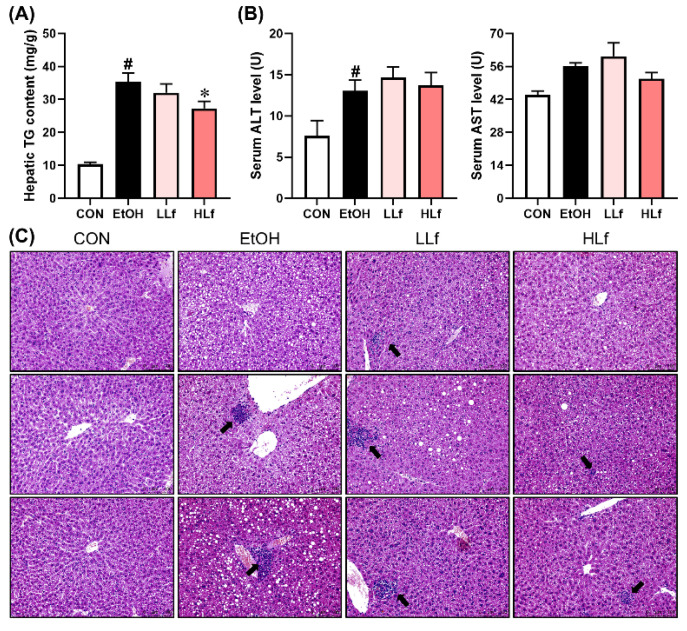
Effects of Lf on hepatic injury. (**A**) HLf decreased hepatic triglyceride content. (**B**) Effects of Lf on serum transaminase levels. (**C**) Representative morphological images of the livers with HE staining for three mice in each group (200×). The black arrow indicates necrosis. CON, control group; EtOH, ethanol administration group; LLf, low-dose lactoferrin group; and HLf, high-dose lactoferrin group. Data are displayed as “mean ± SE”. *n* ≥ 6, for CON group; *n* ≥ 10, for EtOH, LLf, and HLf groups. ^#^ EtOH vs. CON, *p* < 0.05; * LLf or HLf vs. EtOH, *p* < 0.05.

**Figure 3 antioxidants-11-01508-f003:**
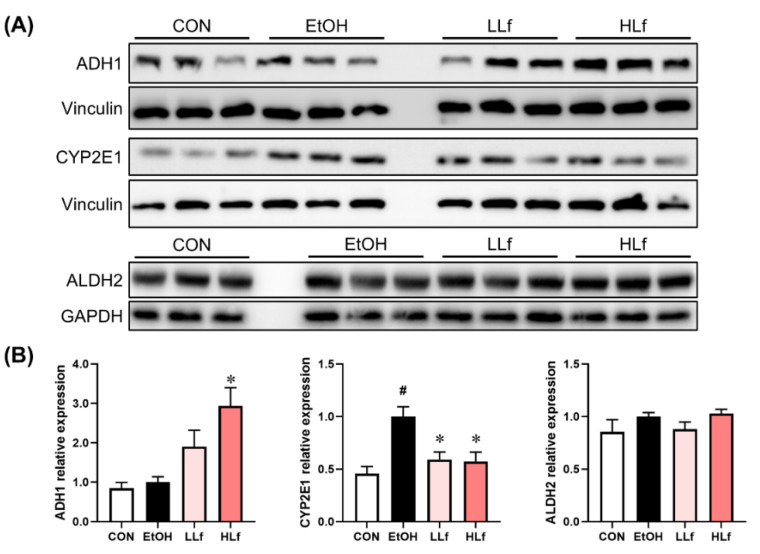
Effects of Lf on the key enzymes of alcohol metabolism in the liver. (**A**) Representative Western blot images. (**B**) Relative expression levels of the alcohol metabolism key proteins. CON, control group; EtOH, ethanol administration group; LLf, low-dose lactoferrin group; and HLf, high-dose lactoferrin group. Data are displayed as “mean ± SE”. *n* ≥ 6, for each group. ^#^ EtOH vs. CON, *p* < 0.05; * LLf or HLf vs. EtOH, *p* < 0.05.

**Figure 4 antioxidants-11-01508-f004:**
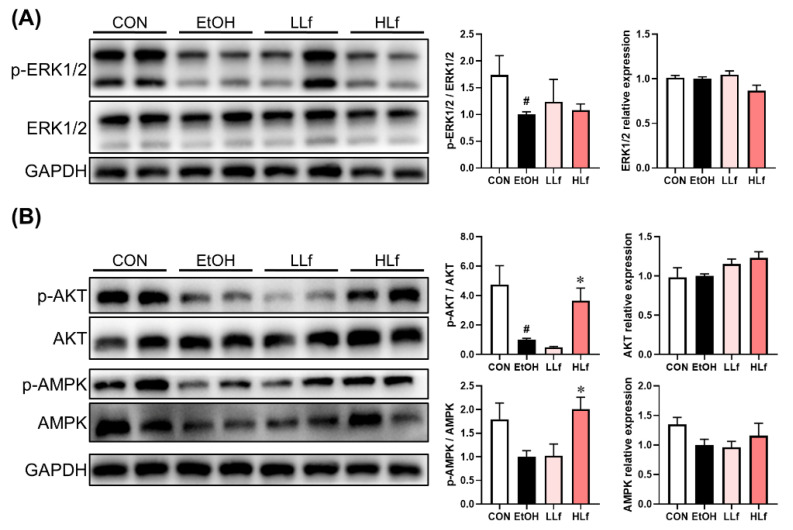
Effects of Lf on protein expressions in redox-sensitive signaling pathways. (**A**) Effects of Lf on ERK1/2 protein expressions. (**B**) Effects of Lf on AKT and AMPK total protein and phosphorylation.CON, control group; EtOH, ethanol administration group; LLf, low-dose lactoferrin group; and HLf, high-dose lactoferrin group. Data are displayed as “mean ± SE”. *n* ≥ 6, for each group. ^#^ EtOH vs. CON, *p* < 0.05; * LLf or HLf vs. EtOH, *p* < 0.05.

**Figure 5 antioxidants-11-01508-f005:**
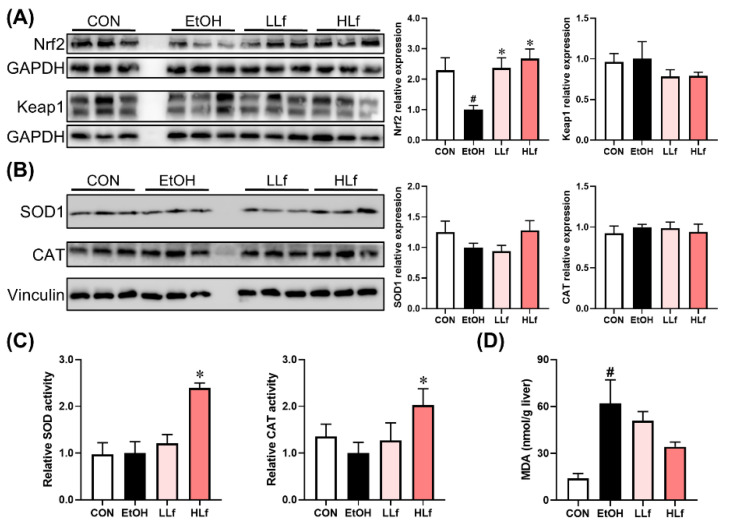
Effects of Lf on redox balance regulatory indicators. (**A**) Effects of Lf on Nrf2 and Keap1 protein expression levels. (**B**) Effects of Lf on hepatic antioxidase activities. (**C**) Effects of Lf on hepatic MDA contents. (**D**) CON, control group; EtOH, ethanol administration group; LLf, low-dose lactoferrin group; and HLf, high-dose lactoferrin group. Data are presented as “mean ± SE”. *n* ≥ 6, for each group. ^#^ EtOH vs. CON, *p* < 0.05; * LLf or HLf vs. EtOH, *p* < 0.05.

**Figure 6 antioxidants-11-01508-f006:**
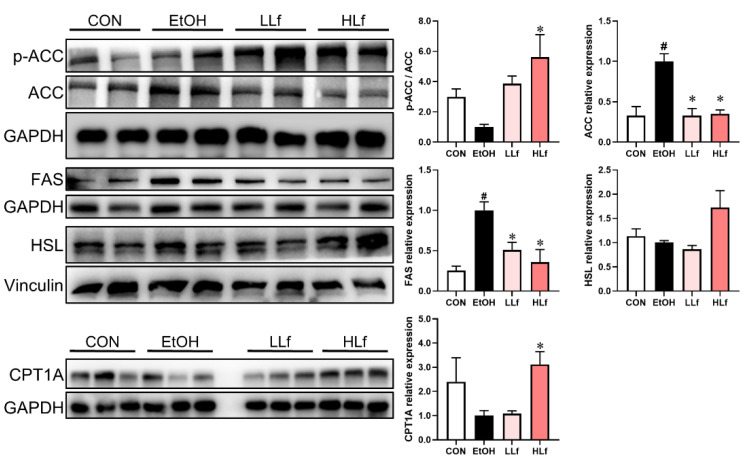
Effects of Lf on lipid metabolism key enzymes at protein level in liver. CON, control group; EtOH, ethanol administration group; LLf, low-dose lactoferrin group; and HLf, high-dose lactoferrin group. Data are presented as “mean ± SE”. *n* = 6, for each group. ^#^ EtOH vs. CON, *p* < 0.05; * LLf or HLf vs. EtOH, *p* < 0.05.

**Figure 7 antioxidants-11-01508-f007:**
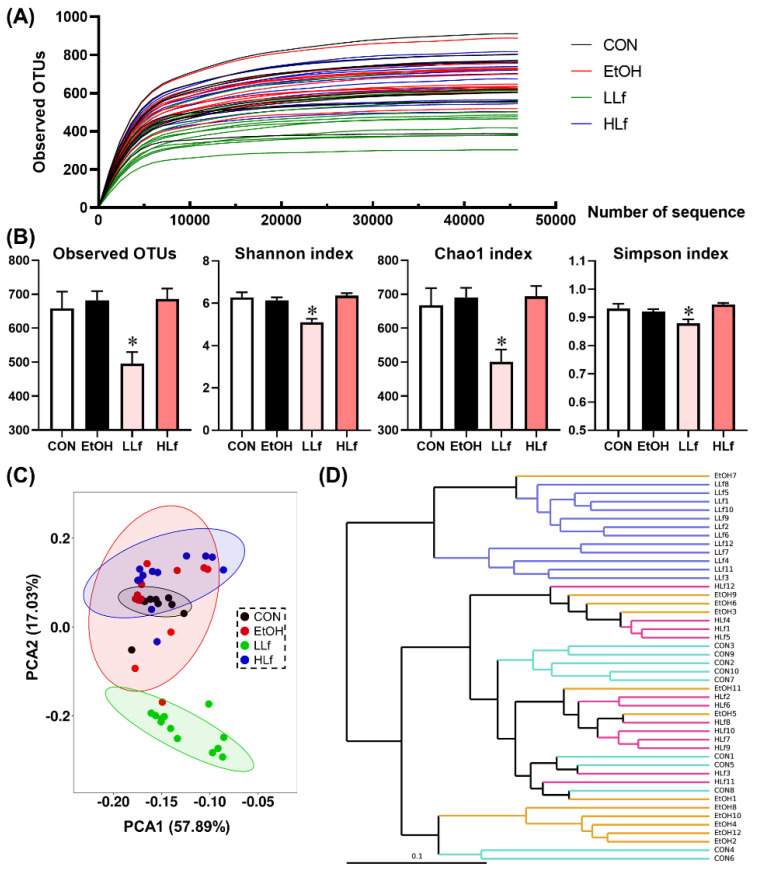
Effects of Lf treatment on alpha and beta diversity of gut microbiota. (**A**) Rarefaction curves; (**B**) alpha diversity of gut microbiota; (**C**) principal component analysis plot; (**D**) UPGMA tree. CON, control group; EtOH, ethanol administration group; LLf, low-dose lactoferrin group; HLf, high-dose lactoferrin group; UPGMA, unweighted pair group method with arithmetic mean. *n* = 10, for CON group; *n* = 12, for EtOH, LLf, and HLf groups. * LLf or HLf vs. EtOH, *p* < 0.05.

**Figure 8 antioxidants-11-01508-f008:**
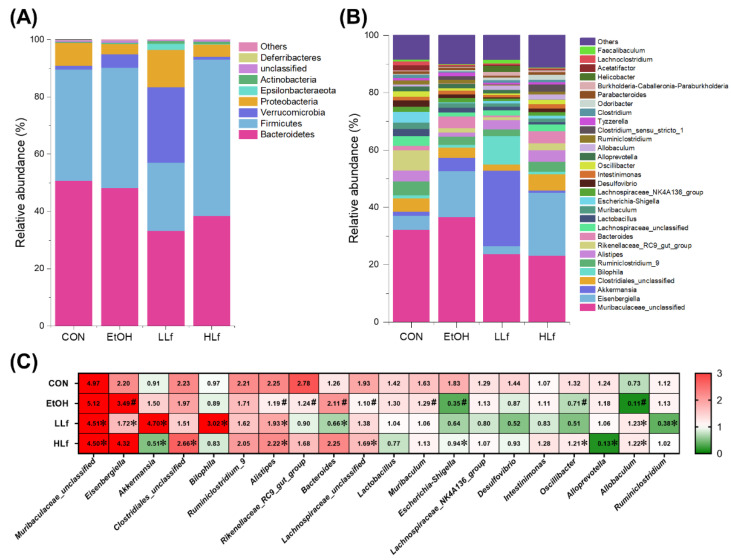
Effects of Lf treatment on gut microbiota relative abundances. (**A**) Changes of relative abundances of gut microbiota at the phylum level. (**B**) Changes of relative abundances of gut microbiota at the genus level. (**C**) Between-group differential analysis at the genus level. Values are presented as “mean for log_2_(relative abundance + 1)” in panel C. *n* = 10, for CON group; *n* = 12, for EtOH, LLf, and HLf groups. ^#^ EtOH vs. CON, *p* < 0.05; * LLf or HLf vs. EtOH, *p* < 0.05.

**Figure 9 antioxidants-11-01508-f009:**
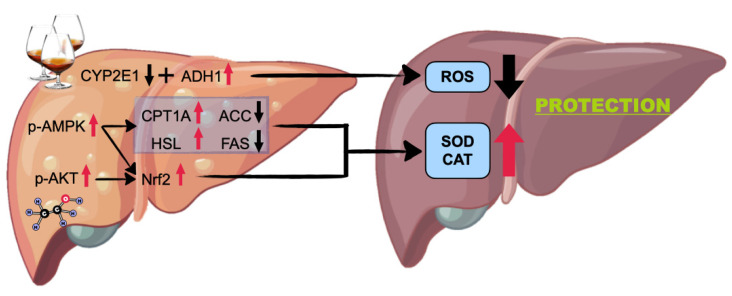
Potential mechanisms for the prevention of high-dose Lf to acute alcoholic liver injury.

**Table 1 antioxidants-11-01508-t001:** Antibody information.

Antibody	Manufacturer	Catalog Number	Country	Dilution
ADH1	CST	5295S	USA	1/1000
CYP2E1	Abcam	28146	UK	1/5000
ALDH2	Proteintech	15310-1-AP	CN	1/2000
*p*-ERK1/2	CST	4370	USA	1/2000
ERK1/2	CST	9102	USA	1/2000
*p*-AKT	CST	4060S	USA	1/2000
AKT	CST	4691S	USA	1/2000
*p*-AMPK	CST	2535	USA	1/1000
AMPK	Santa Cruz	74461	USA	1/200
Nrf2	Proteintech	16396-1-AP	CN	1/1000
Keap1	Proteintech	10503-2-AP	CN	1/5000
SOD1	ABclonal	A0274	CN	1/2000
CAT	ABclonal	A11780	CN	1/2000
ACC	CST	3662S	USA	1/1000
CPT1A	Proteintech	15184-1-AP	CN	1/2000
FAS	ABclonal	A0461	CN	1/1000
HSL	Proteintech	17333-1-AP	CN	1/2000
GAPDH	Proteintech	10494-1-AP	CN	1/20,000
Vinculin	ABclonal	A2752	CN	1/2000

ADH1, alcohol dehydrogenase-1; CYP2E1, cytochrome P450 2E1; ALDH2, acetaldehyde dehydrogenase-2; *p*-ERK1/2, phospho-ERK1/2; ERK1/2, extracellular-signal-regulated kinase 1/2; *p*-AKT, phosphor-AKT; AKT, protein kinase B; *p*-AMPK, phospho-AMPK; AMPK, AMP-activated protein kinase; Nrf2, nuclear-factor-erythroid-2-related factor 2; Keap1, Kelch-like ECH-associated protein 1; SOD1, superoxide dismutase-1; CAT, catalase; ACC, acetyl-CoA carboxylase; CPT1A, carnitine palmitoyltransferase 1A; FAS, fatty acid synthase; HSL, hormone-sensitive lipase; GAPDH, glyceraldehyde-3-phosphate dehydrogenase.

## Data Availability

The data presented in this study are available on request from the corresponding author.

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
