# Peer review of "Lactoferrin Prevents Chronic Alcoholic Injury by Regulating Redox Balance and Lipid Metabolism in Female C57BL/6J Mice"

_antioxidants, 2022, doi:10.3390/antiox11081508_

Round 1
Reviewer 1 Report
In this paper the authors investigate the effects of lactoferrin supplementation (high and low dose, HLf and LLf) in female mice exposed to chronic alcohol consumption.
My main concern is about the statistical analysis that the authors used. They perform ANOVA analysis but then they use the Fisher’s least significant difference (LSD) as a pots-hoc test. The LSD test is a set of individual t tests, does not correct for multiple comparisons, and it is rarely recommended. Therefore, I would appreciate that the authors perform again the statistical analysis by using a multiple comparisons test such as Tuckey.
A second concern is the number of animals used for the different experiments performed. Although they have 10-12 mice/group, the figures show experiments performed with a much lower N. The authors must explain why have used different N, and in that case how have been the samples selected. Moreover, they must clearly indicate the number of samples used, is not enough to say ≥. Finally, in the case of Figure 3, the N=3 is clearly not sufficient.
Other comments:
- Ln 180-190: “HLf treatment also had a more obvious ameliorative effect on steatosis induced by long-term alcohol exposure than LLf treatment”. Is this sentence related to the histology or to the TG measurements? In the first case, the authors should perform an ORO staining of the liver samples to support this affirmation.
- Figure 5: Both LLf and HLf increase NfR2, but only HLf increases SOD and CAT activity. Coulsd the authors speculate why? Maybe it would be useful to examine the amount of these proteins by WB analysis, or alternatively their mRNA abundance.
- Figure 6, and ln 146-148: Regarding ACC, its activity is regulated by phosphorylation. AMPK phosphorylates ACC at Ser79 and this phosphorylation inhibits its enzymatic activity. Despite AMPK was not significantly modified, the authors should determine p-ACC and not only total ACC to be able to conclude about effects on fatty acid synthesis
- Figure 6: The HSL band is not of good quality
Author Response
Response to Reviewer 1 Comments
Ponit 1: My main concern is about the statistical analysis that the authors used. They perform ANOVA analysis but then they use the Fisher’s least significant difference (LSD) as a pots-hoc test. The LSD test is a set of individual t tests, does not correct for multiple comparisons, and it is rarely recommended. Therefore, I would appreciate that the authors perform again the statistical analysis by using a multiple comparisons test such as Tuckey.
Response: Thanks for your comments. We totally agree your opinion. Thus, we re-performed the statistical analysis using Tukey test. Due to the different statistical methods, the P values were changed, so we made appropriate modifications to the corresponding figures and descriptions. However, these changes did not influence our conclusions.
Ponit 2:A second concern is the number of animals used for the different experiments performed. Although they have 10-12 mice/group, the figures show experiments performed with a much lower N. The authors must explain why have used different N, and in that case how have been the samples selected. Moreover, they must clearly indicate the number of samples used, is not enough to say ≥. Finally, in the case of Figure 3, the N=3 is clearly not sufficient.
Response: Thanks for your valuable comments. For the most indicators easy to detect and gut microbiota, we determined the all mice. However, in practical western blot experiment, it is very hard to determine all samples at the same time. Besides, western blots only can measure the relative values but not absolute values, so a direct comparison cannot be conducted between different batches of experiments. Thus, we selected the samples that can reflect population characteristics to perform WB. This is also a common method in animal experiments (PMID: 32780531, 34439499, 31817977, 31463983). Meanwhile, the sample sizes of WB were not completely equal, because loading layout might be different in each WB experiment. Thus, it would have made figure legends lengthy if we reported every N of each panel. We report the lower limit of N. Finally, we agree that N=3 is not sufficient, so we increased the sample size in Figure 3.
Ponit 3:Ln 180-190: “HLf treatment also had a more obvious ameliorative effect on steatosis induced by long-term alcohol exposure than LLf treatment”. Is this sentence related to the histology or to the TG measurements? In the first case, the authors should perform an ORO staining of the liver samples to support this affirmation.
Response: Thanks for your comments. We also agree that ORO staining is superior to HE staining for hepatic steatosis evaluation. In fact, the conclusion depended on both the histology and the TG measurement. On the one hand, we could observe that there are less number and area of fat vacuoles in HLf group than those in LLf group. On the other hand, hepatic TG content is a more direct and precise index to assess liver lipid accumulation, and we found that HLf had a more potent effect on decreasing hepatic TG content. Thus, we claimed “HLf treatment also had a more obvious ameliorative effect on steatosis induced by long-term alcohol exposure than LLf treatment”. The previous descriptions might lead to ambiguity, so we re-wrote the sentence with highlight marker (line 193-195).
Ponit 4:Figure 5: Both LLf and HLf increase NfR2, but only HLf increases SOD and CAT activity. Could the authors speculate why? Maybe it would be useful to examine the amount of these proteins by WB analysis, or alternatively their mRNA abundance.
Response: Thanks. Antioxidase activities are affected by many factors, so an increase of Nrf2 protein level does not necessarily mean the antioxidase activity elevation. We examined the SOD1 and CAT protein expressions by WB according to your suggestions. We found that although LLf and HLf did not up-regulate SOD1 and CAT expressions at protein level, HLf could increase SOD and CAT activity. We speculated that HLf but not LLf can improve SOD and CAT efficiency. Of course, antioxidant enzyme regulation is a complex and multifactorial process. Our speculation may be not perfect, and more research is needed. The relevant contents have been added in the revised manuscript with red font (line 360-365).
Ponit 5:Figure 6, and ln 146-148: Regarding ACC, its activity is regulated by phosphorylation. AMPK phosphorylates ACC at Ser79 and this phosphorylation inhibits its enzymatic activity. Despite AMPK was not significantly modified, the authors should determine p-ACC and not only total ACC to be able to conclude about effects on fatty acid synthesis
Response: Thanks for your comments. We conducted an experiment to supplement these data. Meanwhile, we replace the previous data and figure with the new, and discussed this point in Discussions section with red font in the revised manuscript (line 252-255, 374-377).
Ponit 6:Figure 6: The HSL band is not of good quality
Response: Thanks for your suggestion. We have selected a clearer HSL band to display, and the internal reference protein bond was also replaced accordingly.
Reviewer 2 Report
The manuscript by Li and colleagues is an interesting study about the effect of lactoferrin on a female mouse model of chronic alcohol-mediated liver injury. The authors have published before on the subject using male and female mouse models subjected to chronic/acute alcoholic liver injury. They investigate various metabolic parameters that are known to be affected by alcohol in the liver, together with the expression of enzymes responsible for alcohol metabolism, measuring the influence of two different concentration of lactoferrin on their expression and/or activity. Finally, they attempt a study on the effect of lactoferrin on gut microbiota that has been shown to be regulated by alcohol consumption and lactoferrin itself.
A few questions/remarks that could bring more clarity to the manuscript are the following
Materials and Methods
- There is no need to briefly describe the NGS method, since it is described in detail in Supplementary File 1.
- Placing Figure 1 in the Materials and Methods Section seems inappropriate. Instead, I would place it under section 3.1, where all relevant data are described.
- In the Sample collection paragraph, the authors state that the subject the animals to overnight fasting before sacrificing them the next day, but they do not explain why. Still, in the ERK1/2 Western blot they show an enormous p-ERK1/2 band in the control animals, which could very well be due to fasting, a known inducer of ERK phosphorylation. How do they explain this? There is some ambiguity as to the effect of alcohol in ERK activation, therefore the authors should discuss their result more.
Results
- I would add an introductory sentence in the beginning of Section 3.1 that sets the aim of the described experiment.
- I would add a sentence to point to Figure 1A. Such a sentence is now placed in the Materials and Methods Section.
- I think clarity would be better served if the word “intervention” used by the authors to describe all ethanol-related experiments was changed to “administration”.
Discussion
- In line 323-326, the authors state that ALDH2 is not affected by either dose of lactoferrin because lactoferrin is not potent enough to have a long-lasting effect on ALDH2 protein expression. However, in their previous paper concerning male mice, they showed that Lf induced ALDH2, following 8 weeks of 20% alcohol administration to the animals. Is this a hormonal effect then, or a tolerance effect? Have the authors measured ALDH2 at an earlier time-point than 12 weeks?
- The authors state that the aim of the study is to see if ALI is attenuated by Lf in relation to chronic alcohol administration in females. Since the authors have conducted previous research on male animals they should point out and discuss any obvious differences between the two sexes.
- They conclude that although high dose of lactoferrin has some benefits against ALI following chronic alcohol consumption in female mice, it can have diverse effects in the gut, so Lf supplementation should not be used without medical supervision. Of course, this is a sound advice. However, the authors have not discussed the role/effect of potential lactoferrin supplementation to humans in relation to liver disease, steatohepatitis etc., so this final sentence seems an overreach.
Minor Points
- Line 28: Regulation instead of regulatory.
- Line 193: Figure (B) instead of (F).
- The manuscript should be thoroughly checked for grammatical and syntax errors in English.
Author Response
Response to Reviewer 2 Comments
Ponit 1: There is no need to briefly describe the NGS method, since it is described in detail in Supplementary File 1.
Response: Thanks for your suggestion. We have deleted the descriptions in the revised manuscript.
Ponit 2:Placing Figure 1 in the Materials and Methods Section seems inappropriate. Instead, I would place it under section 3.1, where all relevant data are described.
Response: Thanks. We have placed Figure 1 under section 3.1 according to your suggestions.
Ponit 3:In the Sample collection paragraph, the authors state that the subject the animals to overnight fasting before sacrificing them the next day, but they do not explain why. Still, in the ERK1/2 Western blot they show an enormous p-ERK1/2 band in the control animals, which could very well be due to fasting, a known inducer of ERK phosphorylation. How do they explain this? There is some ambiguity as to the effect of alcohol in ERK activation, therefore the authors should discuss their result more.
Response: Thanks for your comments. Serum ALT and AST determination should be in a fasting condition. Just like people need to fast before physical examination, animals are also fasted before sacrificing in many similar animal experiments (PMID: 21401100, 34116071). We totally agree that ERK1/2 activation in control group might be due to fasting. All mice were fasted, but ERK1/2 phosphorylation seemed to be inhibited in the mice exposed to ethanol. This might suggest that long-term ethanol intake inhibit ERK1/2 activation, even it cannot be restored by fasting, such a known ERK phosphorylation inducer. Of course, the effects of alcohol on ERK still need further study. We can only give reasonable speculation based on the existing literature reports and the current experimental results. In the revised manuscript, we supplemented the related discussions in red font (line 347-352). We hope the work can improve our study.
Ponit 4:I would add an introductory sentence in the beginning of Section 3.1 that sets the aim of the described experiment.
Response: Thanks. We have added a sentence to describe the aim of the experiment in Section 3.1 according to your suggestions.
Ponit 5:I would add a sentence to point to Figure 1A. Such a sentence is now placed in the Materials and Methods Section.
Response: Thanks for your reminding. We also found that the description and position of Figure 1A in the text are not appropriate. Thus, we adjusted it expect the article structure to be more reasonable. The new sentence was highlighted with red font in the revised manuscript (line 84).
Ponit 6:I think clarity would be better served if the word “intervention” used by the authors to describe all ethanol-related experiments was changed to “administration”.
Response: Thanks for your suggestions. We have replaced “intervention” with “administration” when describing ethanol-related experiments.
Ponit 7: In line 323-326, the authors state that ALDH2 is not affected by either dose of lactoferrin because lactoferrin is not potent enough to have a long-lasting effect on ALDH2 protein expression. However, in their previous paper concerning male mice, they showed that Lf induced ALDH2, following 8 weeks of 20% alcohol administration to the animals. Is this a hormonal effect then, or a tolerance effect? Have the authors measured ALDH2 at an earlier time-point than 12 weeks?
Response: Thanks for your comments. Because hepatic ALDH2 determination needed to be conducted after the animals were killed, we did not measure ALDH2 at an early time-point. Our descriptions might be not clear enough. In fact, we used different modelling methods in the present and previous study, so a direct comparison could not be performed. We acknowledged that the evidence is not strong enough to support that “Lf might not have an enough efficacy to maintain a long-term and continuous up-regulation of hepatic ALDH2 expression”. So we re-wrote the contents in the revised manuscript (line 330-334).
Ponit 8: The authors state that the aim of the study is to see if ALI is attenuated by Lf in relation to chronic alcohol administration in females. Since the authors have conducted previous research on male animals they should point out and discuss any obvious differences between the two sexes.
Response: Thanks for your comments. In practical life, male drinkers usually have different drinking behavior with female drinkers. Almost all male drinkers are chronic drinkers, and they often “binge”, however, relatively few women with long-term drinking habits have “binge” behavior. Thus, we used different modeling methods in the present and previous study. Thus, it was hard to conduct a direct comparison between the two sexes. However, we still made a moderate comparison in overlap between two sexes, for example, alcohol metabolism and gut microbiota. The relevant contents have been highlighted in the revised manuscript (line 325-327, 385-394).
Ponit 9: They conclude that although high dose of lactoferrin has some benefits against ALI following chronic alcohol consumption in female mice, it can have diverse effects in the gut, so Lf supplementation should not be used without medical supervision. Of course, this is a sound advice. However, the authors have not discussed the role/effect of potential lactoferrin supplementation to humans in relation to liver disease, steatohepatitis etc., so this final sentence seems an overreach.
Response: Thanks for your comments. The sentence aimed to emphasize that lactoferrin should not be blindly supplemented due to its potential health risk of excessive intake. Our previous descriptions might not be rigorous enough, so we re-wrote the sentence and highlighted it with red font (line 429-430).
Ponit 10: Line 28: Regulation instead of regulatory.
Response: Thanks for your suggestion. “Regulatory” has been changed to “regulation”.
Ponit 11: Line 193: Figure (B) instead of (F).
Response: Sorry for our carelessness. We have changed “Figure F” to “Figure B” in the figure legends.
Ponit 12: The manuscript should be thoroughly checked for grammatical and syntax errors in English.
Response: Thanks for your comments. We have carefully checked and improved the English writing. Please see if the revised manuscript met the English presentation standard.